# Updated cost-effectiveness of MDMA-assisted therapy for the treatment of posttraumatic stress disorder in the United States: Findings from a phase 3 trial

**Elliot Marseille**[1]*, **Jennifer M. Mitchell**[2], **James G. Kahn**[3]

**1** Global Initiative for Psychedelic Science Economics, University of California, Berkeley, CA, United States of America, **2** Department of Neurology, University of California, San Francisco, San Francisco, CA, United States of America, **3** University of California, San Francisco, CA, United States of America

* emarseille1@berkeley.edu

**Data Availability Statement:** All of the cost data are publicly available from the sources mentioned in the article. The outcomes data that are not immediately available via published literature are

## Abstract

### Background

Severe posttraumatic stress disorder (PTSD) is a prevalent and debilitating condition in the United States. and globally. Using pooled efficacy data from six phase 2 trials, therapy using 3,4-methylenedioxymethamphetamine (MDMA) appeared cost-saving from a payer's perspective. This study updates the cost-effectiveness analysis of this novel therapy using data from a new phase 3 trial, including the incremental cost-effectiveness of the more intensive phase 3 regimen compared with the shorter phase 2 regimen.

### Methods

We adapted a previously-published Markov model to portray the costs and health benefits of providing MDMA-assisted therapy (MDMA-AT) to patients with chronic, severe, or extreme PTSD in a recent phase 3 trial, compared with standard care. Inputs were based on trial results and published literature. The trial treated 90 patients with a clinician administered PTSD scale (CAPS-5) total severity score of 35 or greater at baseline, and duration of PTSD symptoms of 6 months or longer. The primary outcome was assessed 8 weeks after the final experimental session. Patients received three 90-minute preparatory psychotherapy sessions, three 8-hour active MDMA or placebo sessions, and nine 90-minute integrative psychotherapy sessions. Our model calculates the per-patient cost of MDMA-AT, net all-cause medical costs, mortality, quality-adjusted life-years (QALYs), and incremental cost-effectiveness ratios (ICERs). We reported results from the U.S. health care payer's perspective for multiple analytic time horizons, (base-case is 30 years), and conducted extensive sensitivity analyses. Costs and QALYs were discounted by 3% annually. Costs were adjusted to 2020 U.S. dollars according to the medical component of the U.S. Bureau of Labor Statistics' Consumer Price Index (CPI).

patient-level CAPS score data that we used to classify trial subjects by PTSD severity categories. The data that support the findings of this study are available from the sponsor (MAPS). However, restrictions apply to the availability of these data, which were used under license for the current study, and so are not publicly available. Those data are available from the Director, Data Management and Services MAPS Public Benefit Corporation, michelle.pleshe@mapsbcorp.com for researchers who meet the criteria for access to confidential data.

**Funding:** The author(s) received no specific funding for this work.

**Competing interests:** The authors have declared that no competing interests exist.

## Results

MDMA-AT as conducted in the phase 3 trial costs $11,537 per patient. Compared to standard of care for 1,000 patients, MDMA-AT generates discounted net health care savings of $132.9 million over 30 years, accruing 4,856 QALYs, and averting 61.4 premature deaths. MDMA-AT breaks even on cost at 3.8 years while delivering 887 QALYs. A third MDMA session generates additional medical savings and health benefits compared with a two-session regimen. Hypothetically assuming no savings in health care costs, MDMA-AT has an ICER of $2,384 per QALY gained.

## Conclusions

MDMA-AT provided to patients with severe or extreme chronic PTSD is cost-saving from a payer's perspective, while delivering substantial clinical benefit.

## Background

Posttraumatic Stress Disorder (PTSD) affects 11.8 million Americans based on past-year prevalence estimates, and a large portion suffer debilitating effects [1]. PTSD is also associated with a wide range of comorbid health conditions. In addition to the direct human toll, PTSD imposes a large economic burden on the country in added health care costs [2] lower productivity, and higher rates of unemployment [3]. Current pharmacological and psychotherapeutic treatments have limited effectiveness for many; and approximately 50% of PTSD patients do not meaningfully respond to standard therapies [4–6]. The need for more effective treatments is widely acknowledged.

Six phase 2 randomized, placebo-controlled, multi-site clinical trials sponsored by the Multidisciplinary Association for Psychedelic Studies (MAPS), tested the safety and efficacy of a novel therapy assisted by 3,4-methylenedioxymethamphetamine (MDMA). A pooled analysis (N = 105) of the results of these trials showed that MDMA-assisted therapy (MDMA-AT), was associated with a 54.2% decrease in patients meeting criteria for PTSD at follow-up. An additional portion exhibited clinically significant improvement, though still met criteria for PTSD. In addition to its encouraging safety and efficacy profile, a cost-effectiveness analysis based on the pooled phase 2 data indicated that this intervention was not only cost-effective, but would save third party healthcare payers $103.2 million (discounted) over 30 years for 1000 patients who meet inclusion criteria. Payers would break even in a little over three years [7].

The recent publication of the first of MAPS' two phase 3 trials added substantial evidence for the benefits of this new therapy [8]. The effectiveness in this trial exceeded the pooled phase 2 results, with 67% of patients no longer meeting criteria for PTSD at the primary endpoint, ~8 weeks following the third experimental session. This economic study re-calculates the cost-effectiveness of MDMA-AT based on the results of this new trial. Cost-effectiveness is important in the coverage deliberations of third party payers, both private insurers and governmental payers such as the Veterans Administration, Medicare, and Medicaid. In view of the likely approval by the U.S. Food and Drug Administration (FDA) of MDMA as a treatment for PTSD in a psychotherapeutic context, it is useful to plan for this eventuality. By reassessing cost-effectiveness, this article is intended to support the insurance and policy planning process.

## Methods

### Overview

We adapted a previously published decision analytic model to portray clinical benefits, MDMA-AT costs, and net medical costs in a hypothetical cohort of 1,000 patients reflecting the distribution by PTSD severity of 90 patients treated in the MAPS-sponsored randomized, placebo-controlled, multi-site phase 3 trial, enrolled between November 2018 and May, 2020 [8]. Patients were portrayed in a Markov process as asymptomatic, or suffering from mild, moderate, severe, or extreme PTSD. MDMA-AT efficacy is portrayed as the change in the distribution of patients by severity category at the trial's endpoint compared with baseline. Mortality by PTSD severity category and medical costs were estimated from published literature. Health state utilities were calculated from EQ-5D-DL surveys completed by trial participants. The Markov simulation is annual for 30 years, costs and QALYs are discounted at 3% annually, and results presented for several time horizons. Costs were adjusted to 2020 using the Bureau of Labor Statistic's medical care Consumer Price Index [9]. Model input values are summarized in Table 1.

### Patient population

The 90 subjects (44 placebo; 46 MDMA) had an average duration of PTSD of 14.1 years (SD, 11.5) and a mean Clinician-Administered PTSD Scale (CAPS-5) total score of 44.1 (SD, 6.04), where "severe" is defined as a total score of 35–47 [10]. PTSD severity of all subjects was severe or extreme at baseline. (See eMethods A on classification of PTSD severity by CAPS-5 score.) The mean age was 41.0 years (SD, 11.9). Females constituted 59.0% and whites 69.0% of the subjects. These patients had received a heterogeneous mix of treatments for PTSD prior to enrollment in the trial. Of the two FDA-approved Selective Serotonin Reuptake Inhibitors (SSRIs) for the treatment of PTSD, 18.9% and 6.7% were on sertraline and paroxetine respectively, prior to baseline assessment and the first experimental session. With regard to non-pharmacological interventions, 37.8% had participated in Cognitive Behavioral Therapy (CBT), 33.3% in Eye movement Desensitization and Reprocessing (EMDR), 36.7% in group therapy 1.1% in prolonged exposure therapy, 23.3% in psychodynamic therapy, and 87.8% had participated in some other form of therapy. Only 2.2% had not participated in any previous therapy [8].

### Treatment protocol

Following recruitment, randomization, and three non-drug 90-minute therapy sessions, participants received blinded doses of an inert placebo or active doses of MDMA. Three 90-minute psychotherapeutic integration sessions were providing following each experimental session. The first was provided on the morning after the experimental session and the remaining two were provided over the following three to four weeks. All sessions were conducted with two therapists formally trained in psychedelic-assisted psychotherapy. Further details on the protocol are provided elsewhere [8].

### Representation of treatment effects

All simulations incorporate the observed outcomes of the phase 3 trial at ~8 weeks following the third experimental session. Because the control condition does not represent a feasible treatment option, we modeled the costs and benefits of the active treatment group after receiving MDMA-AT with the same group at baseline assuming no change in their treatment. Other key features of the model are described in the earlier publication [7] and summarized here:

**Table 1. Model input values, ranges for sensitivity analyses, distributions, and data sources.**

| | Key inputs | Value (+/- range or standard deviation) | Distribution | Source |
|---|---|---|---|---|
| Distribution by PTSD severity: at intake | Asymptomatic | 0 | NA | MAPS phase 3 trial data: Weathers 2011 |
| | Mild | 0 | | |
| | Moderate | 0 | | |
| | Severe | 714 | | |
| | Extreme | 286 | | |
| Distribution by PTSD severity: at primary follow-up | Asymptomatic | 333 | NA | MAPS phase 3 trial data: Weathers 2011 |
| | Mild | 262 | | |
| | Moderate | 262 | | |
| | Severe | 119 | | |
| | Extreme | 24 | | |
| Intervention costs | Therapists | $9,828 | N/A | CMS; Fair Health; MAPS accounting data |
| | Initial screen, lab at intake | $909 | | |
| | Test kits, nuclear stress test, carotid ultrasound, pharmacy | $800 | | |
| | Total | $11,537 (+/- 30%) | Beta | |
| Health care cost (annual) | Asymptomatic | $5,032 ($712) | Gamma | Ivanova 2011, Marciniak 2005, Chan 2009, Lavelle 2018 and authors' construction |
| | Mild | $10,118 ($1,431) | | |
| | Moderate | $15,177 ($2,146) | | |
| | Severe | $20,236 ($2,862) | | |
| | Extreme | $24,283 ($3,434) | | |
| Mortality, relative risk | Asymptomatic | 1.00 | Lognormal | Ahmadi 2011 and authors' construction |
| | Mild | 1.74 (0.70) | | |
| | Moderate | 2.05 (0.80) | | |
| | Severe | 2.51 (1.10) | | |
| | Extreme | 2.76 (1.25) | | |
| Utilities | Asymptomatic | 0.90 (+/- 10%) | Beta | Calculated from EQ-5D-5L data collected in the phase 3 trial. |
| | Mild | 0.83 (+/- 10%) | | |
| | Moderate | 0.74 (+/- 10%) | | |
| | Severe | 0.61 (+/- 10%) | | |
| | Extreme | 0.37 (+/- 10%) | | |
| Other inputs | Cohort size | 1000 | N/A | N/A |
| | Risk of progression (annual) after year 5 | 10% (5% - 15%) | Uniform | Authors' construction; Only used in scenario analysis. |
| | Background mortality | 0.0020 at age 41 | N/A | U.S. Life-tables; National Vital Statistics Reports |
| | Discount rate | 3.0% (0.34%) | Beta | World Bank, 1993 |
| | Time horizon | 5 | N/A | Authors' construction |
| | Mean age | 41 (11.9) | Normal | Mitchell, 2021 |

Patients transitioned to death according to the relative risk of mortality by severity category. In the base-case analysis, patients in both control and treatment conditions remain in the same category achieved at the trial's end. Thus, the intervention effect is captured as the difference between baseline and follow-up in the distribution of patients by severity categories.

This approach comports with a follow-up study of 19 subjects who completed the phase 2 trials and showed a modest, statistically insignificant *improvement* in PTSD at 45.4 months (SD,17.3) [11]. No improvement or remission is also consistent with the stable clinical status of the patients in this trial who suffered from PTSD for an average of 14.1 years at intake [8]. Further, the limited data available on the long-term trajectory of PTSD suggest that

spontaneous remission is predominantly confined to a few years following diagnosis; and after seven years, the 50% of patients without remission experience chronic PTSD [12–14].

The model was implemented in Excel® (Office 365, Microsoft Corporation) and used @RISK® (Palisade Corporation, version 8.1.1) for sensitivity analyses. Costs and QALYs were discounted at 3% per year. MDMA-AT and medical care costs were adjusted to 2020 prices using the Bureau of Labor Statistic's medical care Consumer Price Index [9].

### Health state utility values

The EQ-5D-5L (EuroQol) questionnaire was administered to the phase 3 (but not phase 2) trial participants, and scores were converted to utilities as used in QALY assessments according to the methods outlined in a 2019 study designed specifically to convert EQ-5D-5L scores from the United States into utility measures [15]. Adverse events associated with the trial were transient [8] and are not reflected in the current analysis.

### Intervention costs

Using micro-costing to obtain service delivery costs, Current Procedural Terminology codes were assigned to each MDMA-AT activity and costed according to the average in-provider-network cost in eight metropolitan areas provided by *FAIR Health Consumer* [16] or by Medicare allowable reimbursement [17]. Details are provided in eMethods B. MAPS' accounting data furnished estimates of costs for pharmacy, and tests for pregnancy and drugs of abuse at screening.

### Medical care costs

PTSD is associated with higher mental health and general medical care costs, and costs increase with severity [18]. By applying the medical care component of the Consumer Price Index (CPI) for 2020 [19], we updated our previous estimate of all-cause medical costs borne by U.S. medical care payers for patients with PTSD [7]. The model conservatively distributes the reduction in health care costs over five years: no cost reduction in the first year following MDMA-AT, and 25% of the reduction in each of the four successive years.

### Mortality, analytic time horizon, and outcomes

Our treatment of relative mortality risk, analytic time horizons, and the specification of outcomes are unchanged from the previous analysis and are summarized here: Using the age-specific background U.S. mortality rate as a referent, we set the relative mortality risks at 1.0, 1.74, 2.05, 2.51 and 2.76 for asymptomatic, mild, moderate, severe, and extreme PTSD, respectively. The base-case analysis projects costs and health consequences to 30 years. We also report on results at 1 and 10 years and find the duration of benefits at which net payer costs break-even. We report the following outcomes: Per-patient cost of MDMA-AT; net costs or savings to the health care payer (MDMA-AT cost adjusted for discounted future medical care costs); premature deaths averted; and QALYs gained. In scenarios that do not generate net savings, we calculate the cost per QALY gained.

### Sensitivity and scenario analyses

We conducted one-way and multi-way sensitivity analyses to assess variation in findings given parameter value uncertainty. Sensitivity ranges for the one-way deterministic analyses were informed by low and high estimates (typically 95% confidence intervals) reported in relevant literature. For probabilistic, multi-way sensitivity analyses, we ran 10,000 Monte Carlo

simulations with beta distributions specified for probabilities, gamma distributions for costs, and lognormal distributions for relative risks. We specified distribution parameters such that the central tendencies approximate those reported in the source literature when such information was available. Variables included in the simulations were MDMA-AT cost, health care costs, relative mortality risk and utilities, mean patient age and the discount rate.

Because the long-term durability of MDMA-AT benefits is unknown, in scenario analyses, we portrayed results assuming that five years after trial completion, patients in the MDMA arm, but not in the control arm, progress to the next most severe PTSD category at a rate of 10% and at 20% annually. Finally, we calculated an ICER comparing the cost and health consequences of the less intensive and less costly phase 2 protocol featuring two MDMA sessions per patient, with the more intensive phase 3 protocol which includes three MDMA sessions. To ensure comparability, we applied the updated MDMA-AT costs per CPT code, utility estimates and average patient age of the phase 3 analysis, to both.

## Results

### Base-case

**Intervention cost.** The cost of the MDMA-AT intervention was $11,537 per patient, of which clinicians' compensation constituted 90.7%. 48.7% is attributed to the active MDMA sessions; 27.4% to the 'post' active session integration sessions; 9.1% is compensation for the three sessions that precede the first active session; and 5.6% pertains to clinician time for the screening and intake procedure, adjusted for the 57.3% of patients who were screened out. The remaining 9.3% of costs reflect pharmacy, test kits, laboratory and nuclear stress tests for the 10% of patients who require them. See Fig 1.

**Net costs, QALYs gained, deaths averted and cost-effectiveness.** Projected for 30 years, in a cohort of 1,000 patients MDMA-AT averted 61.4 undiscounted deaths (90% CI, 9.0, 119.3); generated 4,856 discounted QALYs (90% CI, 3,821, 5,679); and saves a discounted $132.9 million (90% CI, $59 - $196 million) in combined mental health and general medical care costs. As shown in Table 2, MDMA-AT is dominant (better and cheaper) unless it is assumed that benefits abruptly cease one year following MDMA-AT. In that case, the

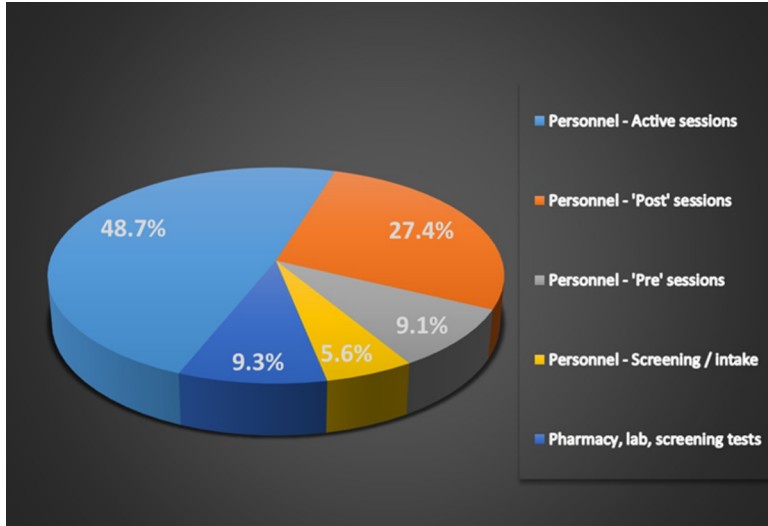

**Fig 1. MDMA-assisted therapy costs by component; Total: $11,537 per patient.**

incremental cost effectiveness ratio (ICER) is $47,554 per QALY gained. Using a 10-year hori-zon, MDMA-AT generates 2,163 QALYs, averts 27.5 deaths, and saves $46.6 million compared with standard of care. MDMA-AT breaks even in costs at 3.8 years, at which point it generates 887 QALYs and averts 10.8 deaths.

### Sensitivity and scenario analyses

In one-way sensitivity analyses, net costs are importantly affected by the mean age of the patient cohort, followed by the cost of treating severe and extreme PTSD, and by the relative mortality risk associated with severe and extreme PTSD. The cost of MDMA-AT has a rela-tively small effect on net savings. See Fig 2. All else equal, because the cost of the intervention is small in comparison with the cumulative potential health care costs savings, a 10% increase in the cost of MDMA-AT causes net discounted 30-year savings to decline by 0.9% (not shown).

MDMA-AT ceases to yield net savings if the added medical care costs associated with PTSD are assumed to be 22.3% of the base case values. In an additional hypothetical (and pes-simistic) scenario in which MDMA-AT has no effect on PTSD-associated medical care costs, it would have an ICER of $2,376 or $5,333 per QALY gained, calculated over 30-year and 10-year horizons, respectively.

Fig 3 shows the results of a 10,000 iteration Monte Carlo scenario analysis which, in addi-tion to variation in the other variables shown in Table 1, allows for annual risk of progression to the next more severe category of PTSD to range from 0.05–0.15 (uniform distribution) after year 5, for the MDMA-AT cohort only. Mean net savings are $69.8 million (90% CI, $125 mil-lion—$20 million), and 3,018 QALYs are gained (90% CI, 2,036–4,057).

### Estimate of the incremental cost-effectiveness of three versus two MDMA sessions

The additional cost of the third 8-hour session plus three accompanying 90-minute integration sessions is $3,088. If MDMA-AT effects were to cease after 12 months, the phase 3 protocol generates 26 additional QALYs at an incremental net cost of $1.8 million for 1,000 patients. The ICER in that case is $66,704 per QALY gained and is 'cost-effective'. In each of the longer analytic time horizons, the comparative benefits of the phase 3 protocol become progressively greater, and is the dominant option. At 30 years it generates an additional 608 QALYs and saves $32.0 million in health care costs compared with the phase 2, two-MDMA session regi-men. See Table 3.

## Discussion

This study of the cost-effectiveness of MDMA-AT provided to severely-affected PTSD patients using recently-published phase 3 trial data strengthens the broad conclusion of an earlier arti-cle based on pooled phase 2 data: Access to MDMA-AT by this group of patients would save the health care system money while generating substantial health benefits [7]. Over 30 years, savings would approximate a discounted $132.9 million while averting 61.4 premature deaths and generating 4,856 discounted QALYs.

The results of the two studies vary in consequential ways. Most importantly, the clinical benefit of the phase 3 trial surpassed that of the phase 2 trials: At the primary endpoint, 14.3% of patients in the MDMA arm of the phase 3 trial were classified as having severe or extreme PTSD, in contrast to 37.8% of the same group of patients in the phase 2 trials. This is the pri-mary reason the economic results from the current study are more favorable; savings of $132.9

**Table 2. Net present costs, health benefits and cost-effectiveness results for 30, 10, 3.8 and 1-year analytic time horizons for 1,000 patients.**

| | | | MDMA-AT | Control |
|---|---|---|---|---|
| **Intervention and future medical care costs** | 30 years | Costs | $234,636,667 | $367,553,320 |
| | | Net cost (savings) | **($132,916,653)** | |
| | 10 years | Costs | $128,711,983 | $175,269,817 |
| | | Net cost (savings) | **($46,557,834)** | |
| | 3.8 years[1] | Costs | 61,210,801 | 61,210,801 |
| | | Net cost (savings) | **$0** | |
| | 1-year | Costs | $32,254,058 | $20,625,082 |
| | | Net cost (savings) | **$11,628,976** | |
| **Health benefits** | 30 years | QALYs | 14,179 | 9,322 |
| | | QALYs gained | **4,856** | |
| | | Deaths | 282.0 | 343.4 |
| | | Deaths averted[2] | **61.4** | |
| | 10 years | QALYs | 6,603 | 4,440 |
| | | QALYs gained | **2,163** | |
| | | Deaths | 50.7 | 78.2 |
| | | Deaths averted[2] | **27.5** | |
| | 3.8 years[1] | QALYs | 2,757 | 1,870 |
| | | QALYs gained | **887** | |
| | | Deaths | 16.1 | 26.9 |
| | | Deaths averted[2] | **10.8** | |
| | 1-year | QALYs | 767 | 522 |
| | | QALYs gained | **245** | |
| | | Deaths | 4.0 | 6.9 |
| | | Deaths averted[2] | **2.9** | |
| **Cost-effectiveness** | 30 years | Net cost per QALY gained | Dominant[3] | |
| | 10 years | | Dominant[3] | |
| | 3.8 years[1] | | Dominant[3] | |
| | 1-year | | **$47,554** | |

Costs and QALYs discounted at 3% annually. MDMA-AT: MDMA assisted therapy; QALY: Quality-Adjusted Life-Year. 1. Analytic horizon at which net costs are zero; 'break-even'; 2. Undiscounted; 3. MDMA-AT is less costly and yields more QALYs; no cost-effectiveness ratio calculated.

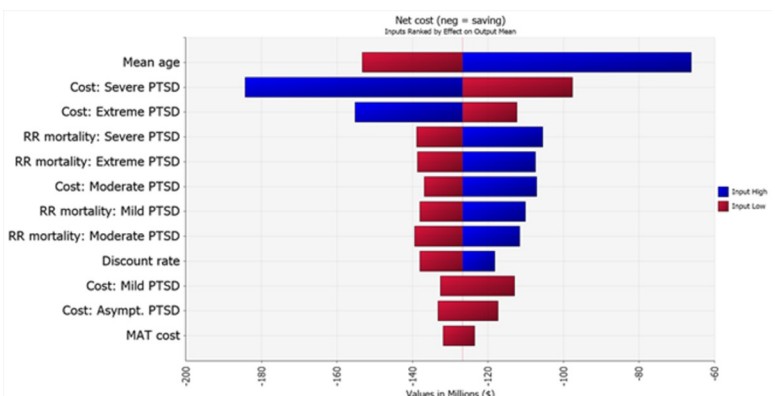

**Fig 2. One-way sensitivity analyses.** Net cost (neg = savings) of MDMA-AT over 30-year analytic time horizon; for 1,000 patients using input variables across the range of values shown in Table 1; Inputs ranked by effect on output mean.

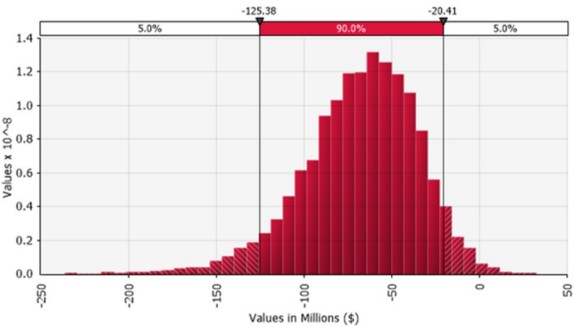

**Fig 3. Multivariable sensitivity analysis on net cost (neg. = savings).** 10,000 iterations of a Monte Carlo simulation assuming PTSD progression risk varied from 0.05 to 0.15 annually in a uniform distribution in patients receiving MDMA-AT; 1,000 patients over a 30-year analytic horizon.

million versus $103.2 million, in spite of the higher cost of the MDMA-AT intervention. The number of averted deaths is also higher in the current study, 61.4 versus 42.9.

However, the overall estimated health benefits over 30 years measured in QALYs was higher in the previous study, 5,553 versus 4,856. This paradoxical-seeming result is due to the improved methods of measuring health-related utility employed in the current study. In the previous analysis, (based on the phase 2 results), the best available estimates of utility associated with PTSD were not well-stratified by severity, and were derived from disparate and thus imperfectly-applicable study populations. In the phase 3 trial, by contrast, the EQ-5D-5L survey was administered to participants, and we were able to derive utility estimates using a consistently-applied method in the most directly-relevant patient population. This improved method yielded a different result. Specifically, the utilities associated with less severe PTSD were lower. For example, the difference between mild and extreme PTSD in the earlier study was 0.97–0.37 = 0.60. In the current study the difference is 0.83–0.37 = 0.46. Thus, all else equal, fewer added QALYs are associated with the transition from more severe to less severe PTSD in the context of the current phase 3 study.

It is likely that the addition of a third MDMA session is responsible for the greater efficacy seen in the phase 3 trial. Evidence for this can be seen in the large drop in CAPS-5 scores between sessions 2 and 3, from 26.2 to 19.5 [8]. We see a similar proportionate decrease in CAPS-IV scores in the pooled phase 2 analysis among subjects who received a third MDMA session, though this was not a primary outcome due to the cross-over design and lack of control group for the third session [20]. Finally, early evidence of enhanced benefit with three

**Table 3. Incremental costs, QALYs and cost-effectiveness: Phase 3 vs phase 2 regimen for 1,000 patients; 1, 5, 10, and 30-year analytic time horizons for 1,000 patients.**

| | 1-Year horizon | | 5-Year horizon | | 10-Year horizon | | 30-Year horizon | |
|---|---|---|---|---|---|---|---|---|
| | **QALYs** | **Net cost (savings)** | **QALYs** | **Net cost (savings)** | **QALYs** | **Net cost (savings)** | **QALYs** | **Net cost (savings)** |
| Phase 2 results | 218 | $9,866,694 | 1,030[1] | ($6,634,409) | 1,915[1] | ($34,848,804) | 4,248[1] | ($100,866,525) |
| Phase 3 results | 245 | $11,628,976 | 1,158 | ($9,824,430) | 2,163 | ($46,557,834) | 4,856 | ($132,916,653) |
| Difference | 26 | $1,762,282 | 129 | ($3,190,021) | 248 | ($11,709,030) | 608 | ($32,050,128) |
| ICER | $66,704 | | Phase 3 –Dominant[2] | | Phase 3 –Dominant[2] | | Phase 3 –Dominant[2] | |

Costs and QALYs discounted at 3% annually. ICER: incremental cost effectiveness ratio; QALY: Quality-Adjusted Life-Year. 1. To facilitate comparisons, QALYs reflect the same utility values for PTSD used in the present study rather than the values used in the analysis based on the pooled phase 2 results (Marseille 2020); 2. MDMA-AT is less costly and yields more QALYs; no cost-effectiveness ratio calculated.

MDMA sessions was noted in a small randomized pilot study in which a statistically significant difference in CAPS scores was found between MDMA sessions 2 and 3 [21]. Taken together, these findings provide substantial evidence of the incremental benefit and the favorable ICER of the three-session regimen.

The phase 3 trial utilized a centralized assessment core. The dynamic between a participant and study team is complex; thus, the use of a centralized pool of assessors across all phase 3 study sites may have removed subtle biases present in the phase 2 data. An additional strength of this study is the estimation of utilities using EQ-5D-5L data drawn directly from the trial subjects. The sensitivity and scenario analyses may also reinforce confidence in the validity of these findings. For example, implausibly assuming no reduction in health care spending associated with PTSD remission, MDMA-AT would have an attractive ICER of $2,384 per QALY over 30 years. Similarly, data are not yet available on the durability of MDMA benefits beyond 3.5 years. However, in a scenario in which progression to more severe disease averaged 10% annually in the MDMA-AT arm, and no further progression is assumed in the control arm, MDMA-AT remains the dominant option generating savings of $69.8 million, and 3,018 QALYs. Even assuming a 20% annual migration to the next most severe PTSD category, MDMA-AT remains the dominant option. The demographic composition of the trial patients is weighted toward better-educated and white people. Just as caution should be exercised in extending the effectiveness results to the general U.S. population, so should extending the cost-effectiveness results.

This is the first study to estimate MDMA-AT cost-effectiveness using phase 3 data. It reinforces and extends earlier findings that this intervention is likely to save lives and reduce morbidity in patients with severe, chronic PTSD, while reducing health care costs. The results of a second MAPS-sponsored phase 3 trial are expected by mid-2022. If this trial demonstrates comparable benefits, clinicians, policymakers, and those who suffer from PTSD should be able to look forward, following expected approval by the FDA, to the rapid adoption of MDMA-AT as a legal therapy. However, broad access and therefore public health impact will depend on the willingness of private and public health care payers to make MDMA-AT available as a benefit to their members. The evidence that they can do so with confidence is now compelling.

## Supporting information

**S1 File.**
(PDF)

## Acknowledgments

We wish to express our thanks to Dominic Trepel, PhD (Trinity College Dublin and the Global Brain Health Institute) for his guidance in the conversion of EQ-5D-5L data to utilities.

## Author Contributions

**Conceptualization:** Elliot Marseille, James G. Kahn.

**Data curation:** Elliot Marseille, Jennifer M. Mitchell.

**Formal analysis:** Elliot Marseille, Jennifer M. Mitchell, James G. Kahn.

**Investigation:** Elliot Marseille.

**Methodology:** Elliot Marseille, Jennifer M. Mitchell, James G. Kahn.

**Project administration:** Elliot Marseille.

**Software:** Elliot Marseille.

**Supervision:** Elliot Marseille.

**Validation:** Elliot Marseille, Jennifer M. Mitchell, James G. Kahn.

**Visualization:** Elliot Marseille.

**Writing – original draft:** Elliot Marseille.

**Writing – review & editing:** Elliot Marseille, Jennifer M. Mitchell, James G. Kahn.

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
