## [Decision Letter · Decision Letter 0]

5 Nov 2021

PONE-D-21-18785Updated cost-effectiveness of MDMA-assisted therapy for the treatment of PTSD: Findings from a phase 3 trialPLOS ONE

Dear Dr. Marseille,

Thank you for submitting your manuscript to PLOS ONE. After careful consideration, we feel that it has merit but does not fully meet PLOS ONE’s publication criteria as it currently stands. Therefore, we invite you to submit a revised version of the manuscript that addresses the points raised during the review process.

We look forward to receiving your revised manuscript.

Kind regards,

Ismaeel Yunusa, PharmD, PhD

Academic Editor

PLOS ONE

Journal Requirements:

Additional Editor Comments (if provided):

In addition to carefully revising your manuscript based on reviewer comments, please consider the following:

1. Please update the title to capture that the cost-effectiveness analysis is for the US.

2. Kindly ensure that your abstract provides the following information: Perspective of the analysis, the country you are estimating the cost-effectiveness from, time horizon (you mentioned multiple, which time horizon was used for the base case analysis? I see 30 years was used in sensitivity analyses and I am assuming that was the horizon for your base case), discounting rate used, and cost year (are cost estimates based on 2020 dollar value as indicated in the main manuscript?).

3. In table 1, under the column named 'Value,' consider clarifying what you are reporting in parenthesis as some readers without hands-on experience with modeling cost-effectiveness can get confused.

4. All acronyms in tables or figures should be defined in a table footnote or figure legend (caption).

5. Your QALY estimates look too high. Is it common in PTSD treatments? Are you actually implementing life-months x utility or life-years x utility, knowing that 1 Year of Life × 1 Utility = 1 QALY?

6. Ensure that the updated manuscript follows the CHEERS guidelines for reporting.

Reviewers' comments:

Reviewer's Responses to Questions

**Comments to the Author**

1. Is the manuscript technically sound, and do the data support the conclusions?

Reviewer #1: Yes

Reviewer #2: Yes

2. Has the statistical analysis been performed appropriately and rigorously? 

Reviewer #1: Yes

Reviewer #2: Yes

3. Have the authors made all data underlying the findings in their manuscript fully available?

Reviewer #1: Yes

Reviewer #2: Yes

4. Is the manuscript presented in an intelligible fashion and written in standard English?

Reviewer #1: Yes

Reviewer #2: Yes

5. Review Comments to the Author

Reviewer #1: This updated cost-effectiveness analysis expands upon a previously published study demonstrating that MDMA-AT is not only clinically beneficial for people suffering from severe PTSD but also cost-saving from a payer's perspective. The findings from the phase 3 study, in addition to the previous phase 2 study, have important implications for policy reform and insurance/third party payer planning should this type of therapy be approved in the US and other countries in the next few years. The manuscript is written in a clear and concise manner and it was relatively easy to follow the methods and modelling techniques which are also described in detail in the prior related publication. I have only a few minor comments/clarifications:

1. Line 64: "perhaps only 50% of PTSD patients do not meaningfully respond..." The word "perhaps" is vague/subjective and might be better off replaced by "approximately" based on the literature.

2. A few acronyms should be spelled out at first instance: CBT, SSRI, EMDR, CPI

3. Lines 197-199: how was this assumption and rate determined?

4. Table 3 appears to be missing footnotes; should clarify that net cost is savings (in red in brackets)?

5. Line 296: states the QALYs for phase 2 were 5553 however in Table 3 it's 4248.

6. What are the limitations of comparing the different methods to measure health-related utility (i.e., the addition of using the EQ-5D-5L in the updated analysis)?

7. Are there limitations to comparing the outcomes of phase 3 at ~8 weeks vs. phase 2 at 3.5 years?

Reviewer #2: Line 36 has two periods. Grammatical typo.

Line 242. I was not able to locate the data you were specifying about the 0.9% decline. I understand it is not shown, and the sentence above it is more significant to the message. I see the figure- But I believe it is implied. Just because it is mentioned again in the conclusion, perhaps another figure or table? Or further explanation for the calculation? Overall, well constructed work and analysis.

6. PLOS authors have the option to publish the peer review history of their article (what does this mean?). If published, this will include your full peer review and any attached files.

Reviewer #1: **Yes: **Elena Argento

Reviewer #2: **Yes: **Sarah Tedesco

---

## [Author Response · Author response to Decision Letter 0]

11 Jan 2022

PONE-D-21-18785 - Response to reviewers and editors - Nov 10, 2021

Thank you for this opportunity to respond to both editors’ and reviewers’ comments. We believe that the article is stronger now as a result of this review process.

Each comment is listed below with a description of our corresponding revision. We will upload both the “track changes” version and the final version of the manuscript.

Sincerely,

Elliot Marseille 

Response: We have attended to these concerns, and believe that the article now conforms with PLOS ONE’s requirements.

Additional Editor Comments (if provided):

In addition to carefully revising your manuscript based on reviewer comments, please consider the following:

1. Please update the title to capture that the cost-effectiveness analysis is for the US.

Response: Agree. Done.

2. Kindly ensure that your abstract provides the following information: Perspective of the analysis, the country you are estimating the cost-effectiveness from, time horizon (you mentioned multiple, which time horizon was used for the base case analysis? I see 30 years was used in sensitivity analyses and I am assuming that was the horizon for your base case), discounting rate used, and cost year (are cost estimates based on 2020 dollar value as indicated in the main manuscript?).

Response: Agree. Done. In addition to the items mentioned above, we re-ordered the sentences in the Methods section of the Abstract so that it flows more logically now. 

3. In table 1, under the column named 'Value,' consider clarifying what you are reporting in parenthesis as some readers without hands-on experience with modeling cost-effectiveness can get confused.

Response: Agree. Done. We also added this note to both Tables ,“Costs and QALYs discounted at 3% annually”

4. All acronyms in tables or figures should be defined in a table footnote or figure legend (caption).

Response: Agree. Done

5. Your QALY estimates look too high. Is it common in PTSD treatments? Are you actually implementing life-months x utility or life-years x utility, knowing that 1 Year of Life × 1 Utility = 1 QALY?

Response: I think that the perceived problem is that these results are for 1000 patients. The scale is arbitrary but 1000 is often used in CEAs. I mentioned the 1000 patients more prominently in the title of Table 3 now.

6. Ensure that the updated manuscript follows the CHEERS guidelines for reporting.

Response: Agree. Done. 

Reviewer #1: 

This updated cost-effectiveness analysis expands upon a previously published study demonstrating that MDMA-AT is not only clinically beneficial for people suffering from severe PTSD but also cost-saving from a payer's perspective. The findings from the phase 3 study, in addition to the previous phase 2 study, have important implications for policy reform and insurance/third party payer planning should this type of therapy be approved in the US and other countries in the next few years. The manuscript is written in a clear and concise manner and it was relatively easy to follow the methods and modelling techniques which are also described in detail in the prior related publication. I have only a few minor comments/clarifications:

1. Line 64: "perhaps only 50% of PTSD patients do not meaningfully respond..." The word "perhaps" is vague/subjective and might be better off replaced by "approximately" based on the literature.

Response: Agree. Done.

2. A few acronyms should be spelled out at first instance: CBT, SSRI, EMDR, CPI

Response: Agree. Done.

3. Lines 197-199: how was this assumption and rate determined?

Response: The above lines refer to this passage:

Because the long-term durability of MDMA-AT benefits is unknown, in a scenario analysis, we portrayed results assuming that five years after trial completion, patients in the MDMA arm, but not in the control arm, progress to the next most severe PTSD category at a rate of 10% annually.

As noted in the manuscript, limited existing data on long-term benefits suggests stable and even improving outcomes to 3.5 years. The clinical course after that is unknown, but there is no reason to suspect dramatic deterioration of benefit after that time. Confronted with this uncertainty, we arbitrarily chose a substantial reduction of benefit of 10% per year starting in year 6 and varied this from 5% - 15% in sensitivity analyses. This scenario is intended to add another illustration of the insensitivity of our results to assumptions about the duration of benefits. The 10% progression rate implies significantly reduced cumulative benefits from MDMA-AT. For example, for 1,000 patients, 10 years after the trial, under base-case assumptions, 322 patients would be asymptomatic and 22 would have extreme PTSD. In the scenario of accelerated disease progression, only 171 would be asymptomatic and 106 would suffer from extreme PTSD 10 years out; yet MDMA-AT remains the dominant option. To reinforce just how robust this finding is we added this sentence at around line 346 in the Discussion.

Even assuming a 20% annual migration to the next most severe PTSD category, MDMA-AT remains the dominant option.

4. Table 3 appears to be missing footnotes; should clarify that net cost is savings (in red in brackets)?

Response: Agree. Done

5. Line 296: states the QALYs for phase 2 were 5553 however in Table 3 it's 4248.

Response: Good catch – thanks. We added the following note to the table pertaining to “QALYs” columns for the row with the Phase 2 trial results: “To facilitate comparisons, QALYs reflect the same utility values for PTSD used in the present study rather than the values used in the analysis based on the pooled phase 2 results (Marseille 2020)”. 

FYI: in the Discussion we say: 

“However, the overall estimated health benefits over 30 years measured in QALYs was higher in the previous study, 5,553 versus 4,856. This paradoxical-seeming result is due to the improved methods of measuring health-related utility employed in the current study. In the previous analysis, the best available estimates of utility associated with PTSD were not well-stratified by severity, and were derived from disparate study populations.” 

Nevertheless, I think that the added note in the table will avoid potential confusion. 

6. What are the limitations of comparing the different methods to measure health-related utility (i.e., the addition of using the EQ-5D-5L in the updated analysis)?

Response: We are not 100% sure we understand what the issue is here, but let me see if this addresses it: Following on the passage pasted in the quoted passage directly above, in the Discussion section we had written,

In the previous analysis, the best available estimates of utility associated with PTSD were not well-stratified by severity, and were derived from disparate study populations. The phase 3 trial included the EQ-5D-5L survey, and we were able to derive utility estimates using standard methods. However, the utilities associated with less severe PTSD were lower using this more accurate method. For example, we assigned a utility of 1.0 for asymptomatic patients in the previous analysis, whereas the empirical results from the phase 3 study indicated a utility of 0.9. Thus, all else equal, fewer QALYs are associated with the transition from more severe to less severe PTSD.

In the revised draft we edited for clarity as follows,

In the previous analysis (based on the phase 2 results), the best available estimates of utility associated with PTSD were not well-stratified by severity, and were derived from disparate and thus imperfectly-applicable study populations. In the phase 3 trial, by contrast, the EQ-5D-5L survey was administered to participants, and we were able to derive utility estimates using a consistently-applied method in the most directly-relevant patient population. This improved method yielded a different result. Specifically, the utilities associated with less severe PTSD were lower. For example, the difference between mild and extreme PTSD in the earlier study was 0.97 – 0.37 = 0.60. In the current study the difference is 0.83 – 0.37 = 0.46. Thus, all else equal, fewer added QALYs are associated with the transition from more severe to less severe PTSD in the context of the current phase 3 study.

7. Are there limitations to comparing the outcomes of phase 3 at ~8 weeks vs. phase 2 at 3.5 years?

Response: We don't think so. First, the long-term 3.5 year follow-up from the phase 2 trials was conducted in a small (n = 19) subset of patients. While it is encouraging, it cannot be considered definitive. Second, we can think of no reason that the more favorable results from the three-MDMA-session phase 3 trial regimen should be less durable than the results from the less intensive two-MDMA-session phase 2 trials. Third, the sensitivity and scenario analyses show that the favorable cost-effectiveness results are insensitive to a wide range of plausible assumptions about the duration of benefits, including the scenario discussed above in which 10% of MDMA-AT recipients, but no control patients, progress to more severe PTSD each year. To emphasize this point, we changed lines 344-346 to read as follows

However, in a scenario in which progression to more severe disease averaged 10% annually in the MDMA-AT arm, and no further progression is assumed in the control arm, MDMA-AT remains the dominant option, generating savings of $69.8 million, and 3,018 QALYs.

Reviewer #2: 

Line 36 has two periods. Grammatical typo.

Response: Thank you. Corrected.

Line 242. I was not able to locate the data you were specifying about the 0.9% decline. I understand it is not shown, and the sentence above it is more significant to the message. I see the figure- But I believe it is implied. Just because it is mentioned again in the conclusion, perhaps another figure or table? Or further explanation for the calculation? Overall, well constructed work and analysis

Response: Fair point. We edited lines 251- 253 to read, 

All else equal, because the cost of the intervention is small in comparison with the cumulative potential health care costs savings, a 10% increase in the cost of MDMA-AT causes net discounted 30-year savings to decline by 0.9% (not shown). 

To be clear, we do not mention the 0.9% figure in the conclusion, though we do refer to a 0.9 utility estimate.

---

## [Editor Report · Decision Letter 1]

17 Jan 2022

Updated cost-effectiveness of MDMA-assisted therapy for the treatment of PTSD: Findings from a phase 3 trial

PONE-D-21-18785R1

Dear Dr. Marseille,

We’re pleased to inform you that your manuscript has been judged scientifically suitable for publication and will be formally accepted for publication once it meets all outstanding technical requirements.

Kind regards,

Ismaeel Yunusa, PharmD, PhD

Academic Editor

PLOS ONE
---

## [Editor Report · Acceptance letter]

3 Feb 2022

PONE-D-21-18785R1 

 Updated cost-effectiveness of MDMA-assisted therapy for the treatment of posttraumatic stress disorder in the United States: Findings from a phase 3 trial 

Dear Dr. Marseille:

I'm pleased to inform you that your manuscript has been deemed suitable for publication in PLOS ONE. Congratulations! Your manuscript is now with our production department. 

Kind regards, 

on behalf of

Dr. Ismaeel Yunusa 

Academic Editor

PLOS ONE